# Challenges and practices identification in complex outsourcing relationships: A systematic literature review

**Ghulam Murtaza Khan**[1,2], **Siffat Ullah Khan**[1], **Habib Ullah Khan**[3]*, **Muhammad Ilyas**[1]

**1** Department of Computer Science and IT, Software-Engineering-Research-Group (SERG-UOM), University of Malakand, Chakdara, Pakistan, **2** Department of Computer Science, Shaheed Benazir Bhutto University, Sheringal, Dir(U), Pakistan, **3** Department of Accounting and Information Systems, College of Business and Economics, Qatar University, Doha, Qatar

* habib.khan@qu.edu.qa

**Data Availability Statement:** All relevant data are within the paper.

**Funding:** This publication was supported by Qatar National Library, Doha, Qatar and Qatar University Internal Grant No. IRCC- 2021-010. The findings

## Abstract

Complex IT outsourcing relationships aptitude several benefits such as increased cost likelihood and lowered costs, higher scalability and flexibility upon demand. However, by virtue of its complexity, the complex outsourcing typically necessitates the interactions among various stakeholders from diverse regions and cultures, making it significantly more challenging to manage than traditional outsourcing. Furthermore, when compared to other types of outsourcing, complex outsourcing is extremely difficult because it necessitates a variety of control and coordination mechanisms for project management, which proportionally increases the risk of project failure. In order to overcome the failure of projects in complex outsourcing relationships, there is a need of robust systematic research to identify the key challenges and practices in this area. Therefore, this research implements systematic literature review as a research method and works as a pioneer attempt to accomplish the aforementioned objectives. Upon furnishing the SLR results, the authors identified 11 major challenges with 67 practices in hand from a total of 85 papers. Based on these findings, the authors intend to construct a comprehensive framework in the future by incorporating robust methodologies such as AHP and fuzzy logic, among others.

## Introduction

Outsourcing is the transfer of the continuous management responsibility of a client party or service user to a third party, referred to as a vendor or provider, to perform an IT service under a contract of service level agreement [1]. Outsourcing has since become a well-established academic subject because of its rapidly developing practice. Outsourcing determinations, outsourcing methods, challenges identified in outsourcing etc. are some of the hot topics in the literature [2].

Some of the advantages of outsourcing include ensuring maximum profit, lower costs, increased productivity, flexibility in meeting service needs, higher quality, redirecting company resources to core activities, more customer satisfaction, continuity and risk management,

achieved herein are solely the responsibility of the authors.

**Competing interests:** The authors have declared that no competing interests exist.

faster time to market, access to skilled resources, flexibility to focus on key areas, and faster and better services [3, 4].

Proponents of IT outsourcing have underlined the importance of managing outsourced relationships. "A long-term commitment, shared risk and benefits, a sense of reciprocal cooperation, and other features congruent with concepts and theories of participatory decision making" is what a relationship is characterised as. While IT outsourcing interactions have gotten some attention in the literature, just a few academics have looked at the client-supplier IT outsourcing relationship [5].

Collaboration across corporate boundaries is a necessary component of today's company. This type of client-vendor relationship frequently goes beyond the typical contractual restrictions agreed upon at the start of collaboration. The risks and advantages of joint labours, as well as investments and work load, are evenly distributed among the collaborative partners. Inter- and intra-organizational collaboration helps businesses gain a competitive advantage. Bidirectional trust, reciprocal dependency, and a win-win mindset between partners are the foundations of long-term working relationships. Collaboration is typically developed by businesses to reduce the costs of collecting the necessary information/understanding, capabilities, and competencies for well-organized professional operations [6].

All IT outsourcing contracts include components of collaboration, virtual cooperation, and the demands of increasingly sophisticated systems. But one of the most essential variables in deciding the success or failure of virtual collaboration is trust. Good relationships are tactical assets that necessitate the ongoing management effort and focus [7].

An outsourcing project's success or failure is determined by a number of elements, including project size, duration, and, most importantly, contract design and management among different stakeholders. Therefore, additional research is required to understand how various parts of outsourcing interact and what the further consequences of the dynamics of these factors are on various organizational outcomes [8].

Outsourcing has progressed through several stages of client-vendor relationships. Such phases of the relationship are briefly illustrated via the framework in [9] as

1. Dyadic outsourcing relationship

A client in a dyadic relationship relies on only one vendor to meet all of their demands, which might range from simple to complex. The majority of research in the literature considers such relationship as one-to-one, implying that one client seeks services independently of others, and suppliers do the same.

2. Multi-vendor relationship

In a one-to-many relationship, a client seeks out more than one vendor to supply their services in order to meet the client's needs or achieve its goals, and information regarding the division of labor is shared and discussed with all parties to the agreement.

3. Co-sourcing relationship

A many-to-one or co-sourcing relationship is described as a collaboration between multiple clients for the delivery of a service through a single vendor contract.

4. Complex outsourcing relationship

The phrase complex relationship or complex outsourcing relationship (COR), which is the subject of this study, refers to a relationship between several clients and multiple vendors (i.e., many-to-many), in which multiple clients rely on multiple vendors to meet their needs under the same contract.

OR

Multiple vendor organizations may work for multiple client organizations through the same contractual agreement or connection in a "complex outsourcing relationship (COR). It is worth noting that this study's focus is on such a phase or relationship, which is exceptionally

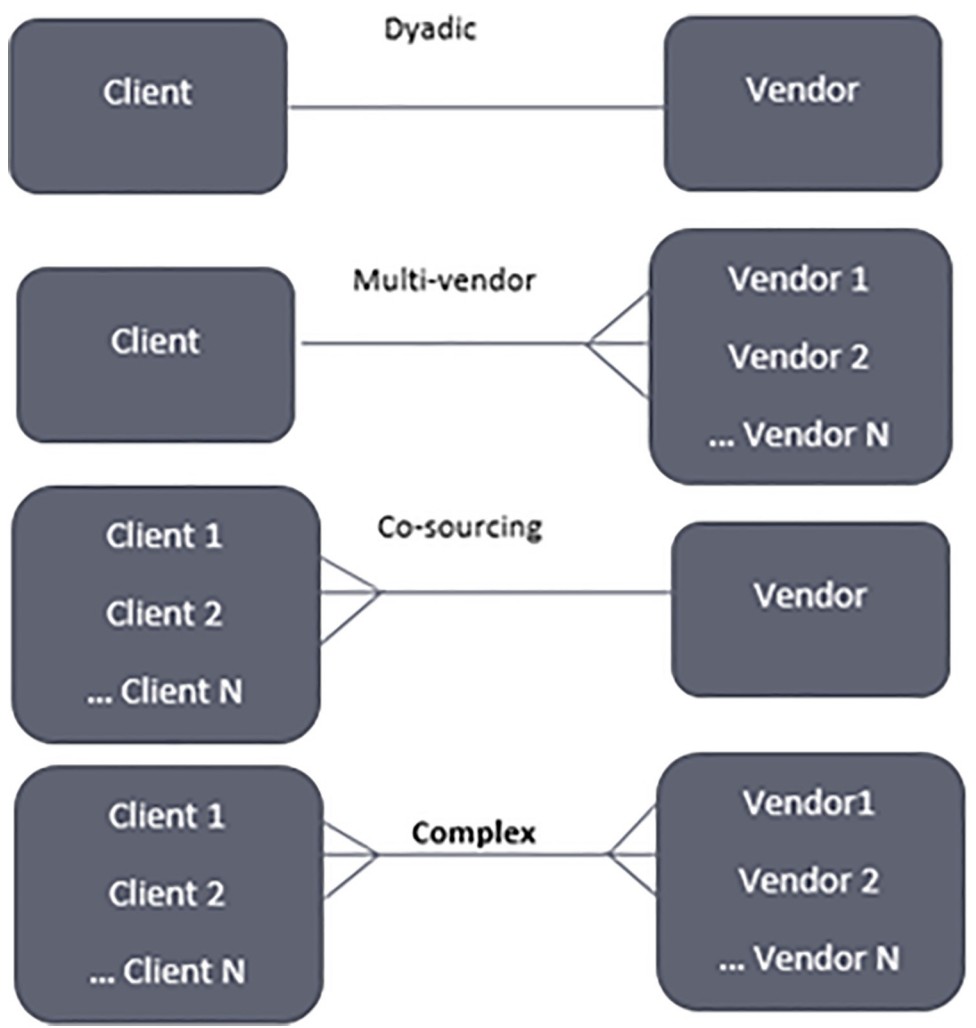

**Fig 1. Types of outsourcing relationships.**

difficult to manage owing to the complexity of the situation and the various types of stakeholders involved. The aforementioned phases and types of outsourcing relationships are shown in (Fig 1).

Good relationships and carefully drafted contracts have a significant impact on outsourcing. However, there is a lack of comprehensive guidance in this area, specifically in the context of complex outsourcing relationships, due to the dearth of studies described in the literature [10]. Outsourcing has become more complex and diversified as a result of the expansion of numerous actors connected via a contract, such as a client and one or more vendors, each with their own unique qualities [11]. Furthermore, because of its complexity, complex outsourcing typically necessitates interactions among various stakeholders from diverse regions and cultures, making it significantly more challenging to manage than traditional outsourcing. Moreover, complex outsourcing when compared to other types of outsourcing is extremely difficult because it necessitates a variety of control and coordination mechanisms for project management, which proportionally increases the risk of project failure [12].

Therefore, in light of these points, this study aims to identify the challenges and practices in the context of complex outsourcing relationships.

Based on this hypothesis, the authors of this study formulated the following research question:

**RQ. 1:** What are the key challenges, as identified in the literature, faced by multiple stakeholders (clients and vendors) in the context of complex outsourcing relationships?

**RQ. 2:** What are the practices/solutions, as identified in the literature, faced by multiple stakeholders (clients and vendors) in the context of complex outsourcing relationships?

The remainder of this paper is structured as follows. Background information is provided in Section II. Section III provides a detailed description of the research methodology. The conclusions of this investigation are presented in section IV. The study's shortcomings are discussed in Section V, and conclusions and future work are discussed in Section VI. The remaining sections provide further information.

## Background

In support of a thorough software, the author proposed [13] a robust architecture for security and compliance challenges. Furthermore, by establishing software requirements based on a study of work patterns of service providers acting in complex IT outsourcing agreements and proposing a software architecture that meets these requirements, this study contributes to filling the gap in software support.

In COR setups, there are a number of solutions [12] that focus on IT infrastructure compliance, but they are typically ineffective and inefficient and do not work well in general. I am not aware of any solutions that support all of the services we use, necessitating the use of multiple solutions.

Complex IS outsourcing initiatives have distinct tasks and stakeholder connections compared to standard IS outsourcing projects [14].

Because of the growing worldwide distribution of IT services, which involves complex interorganizational connections and the dispersion of decision rights and responsibilities across numerous organizations, a thorough understanding of governance procedures is required [15].

The authors build and offer a theoretical framework for explaining various types of outsourcing partnerships, as well as an analysis of why client organizations seeking IT services might choose more collaborative or complicated outsourcing arrangements than dyadic interactions. Moreover, the authors believe that the COR relationships in the future will be more in use [9].

The authors suggested a methodology and decision-making approach [16] for examining incentive schemes and creating outsourcing contracts that benefit both the outsourcer and vendor in COR.

Complex outsourcing contracts [17] are difficult to manage and of themselves, but when time zone differences, language barriers, significant distance, rare face-to-face meetings, and diversity–all of which are common offshore deal companions–the task becomes exponentially more difficult.

Organizations that deal with complex outsourcing arrangements on a regular basis have intelligently implemented the lifecycle model explained in this article [18].

Many companies struggle to properly manage one or more complex outsourcing agreements, generate value, and mitigate risk [19].

This research [20] examines the significance of trust in complex contracts and relationships frameworks that underpin global outsourcing, as well as the difficult conflict-resolution procedures that can be employed to restore trust.

In [21], the authors want to see what an Indian supplier considers to be significant in order to manage some big and complex outsourcing partnerships, and specifically, how reciprocally profitable and long-term partnerships are developed and maintained with European customers.

By offering a descriptive framework incorporating essential governance variables, this article [22] seeks to deliver a good understanding for managing complex-type IT-outsourcing agreements. The findings of the analysis, as well as the framework itself, demonstrate the numerous complex difficulties that arise when managing sophisticated IT outsourcing partnerships.

## Research methodology

We conducted a systematic literature review to collect relevant data from the literature, which is a more complete and rigorous method than the traditional literature review (OLR). The SLR was carried out in three stages: planning, execution, and reporting.

### Search strategy

To conduct an SLR, we tracked Kitchenham's guidelines [23] and Mendes et al. [24]. In addition, depending on the study objectives, we used the population, intervention, comparison, and outcomes (PICO) criteria to select keywords and build search strings.

**Population:** Clients and Vendors in Complex Outsourcing Relationships
**Intervention:** Challenges, Practices
**Comparison:** For the sake of this investigation, no comparisons are made.
**Relevance Outcomes:** To Assist Clients and Vendors in Complex Outsourcing Relationships

Furthermore, to build the search strings, we employed Boolean connectors AND and OR to connect the features of PICO.

### Search strings

To avoid a search break by different database constraints, we developed three different strings.

**String 1:** (("Complex outsourcing" OR "Complex software outsourcing" OR "Complex Information Systems outsourcing" OR "complex IT outsourcing" OR "complex outsourcing relationship") AND (Challenges OR issues OR barriers OR risks OR Practices OR solutions).

**String 2:** (("Complex outsourcing relationship" OR "Complex software outsourcing relationship" OR "Complex Information Systems outsourcing relationship" OR "complex IT outsourcing relationship" OR "Complex outsourcing") AND

(Challenges OR barriers OR issues OR risks OR Practices OR solutions) AND

("Relationship management in complex outsourcing" OR "Co-ordination in complex outsourcing" OR "Communication in complex outsourcing" OR "Contract management in complex outsourcing" OR "Task allocation in complex outsourcing" OR "Trust building in Complex outsourcing"))

**String 3:** (("Complex software outsourcing" OR "Complex IS/IT outsourcing" OR "complex outsourcing") AND ("Challenges" OR "issues" OR "barriers" OR "risks" OR "Practices" OR "solutions" solutions))

### Literature resources

Using our search strings, we applied our search for famous libraries, such as IEEE Xplore, Google Scholar, ACM, Wiley Online Library, SpringerLink, and ScienceDirect. We also used

the snowballing method to further support our research questions and not to miss any important data. It is worth noting that we started the search on 21[th] February, 2021, and systematically completed it on 8[th] April, 2021.

## Criteria for study selection

The strings and authors' recommendations were backed up by the researchers. Initially, we added a string to the library for metadata. The same procedure was followed to avoid interference with the title, abstract, or keyword restrictions. Each paper was properly documented by the first author, who kept a complete record. Based on this phase, other authors assessed the papers and assigned pertinent information for each paper, as well as its title and abstract.

We established the following inclusion and exclusion criteria based on the aforementioned principles for SLR.

**Inclusion criteria.** Such criteria specify which portions of the literature will be considered for inclusion in the selection. Based on such criteria, we have only analyzed items that are relevant to complex outsourcing. The inclusion criteria were as follows:

- Articles in full text of the English language.

- Sources only relevant to COR.

- Studies that describe the COR challenges.

- Studies that describe the COR practices.

- Studies published in journals and conferences.

- White papers and standardized reports from trustworthy organizations.

**Exclusion criteria.** Such criteria specify which pieces of literature are not included for consideration. The exclusion criteria were as follows:

- Studies, not relevant to our research questions.

- Studies of other than English language.

- Incomplete Studies.

- Duplicate studies

- Thesis or magazine and/or web articles.

The search outcomes are presented in Table 1. We retrieved 281 publications from a total of 1372 using the inclusion criteria. Using the exclusion criteria, we narrowed it to 85 papers.

**Table 1. Search strings outcomes documentation.**

| Search Stings | Libraries | Total Articles | Initial-Selection | Final-Selection |
|---|---|---|---|---|
| String 1 | IEEE Xplore | 257 | 25 | 10 |
| | Google Scholar | 870 | 179 | 58 |
| | ACM | 11 | 09 | 02 |
| | Wiley Online Libray | 24 | 19 | 05 |
| String 2 | SpringerLink | 116 | 03 | 01 |
| String 3 | ScienceDirect | 67 | 19 | 03 |
| | Snowballing | 27 | 27 | 06 |
| **Total** | | **1372** | **281** | **85** |

**Table 2. Quality based assessment criteria.**

| ID | Quality Criteria | Yes (1) | Partly (0.5) | No (0) |
|---|---|---|---|---|
| Q.1 | Is the study properly defined in terms of its goals and objectives? | | | |
| Q.2 | Is there a clear context for the item such as a workplace or a laboratory? | | | |
| Q.3 | Is the paper clear about the limitations? | | | |
| Q.4 | Does the paper present any challenge or challenges in the context of COR? | | | |
| Q.5 | Does the paper present any practice or practices in the context of COR? | | | |

During data extraction, the principal author finished each step, which was then examined by other authors.

The primary goal of quality assessment was to identify and eliminate lower-quality studies and to determine the validity of a study's conclusions.

## Criteria for quality assessment

We created the assessment criteria using the guidelines and updates for the SLR from previous studies. In this way, we formulated the questions given in Table 2 prior to the implementation of such criteria. Furthermore, we employed a three-tiered scale to rate each question in the reviewed papers. Yes, no, or partial. We assigned values of 1 to Yes, 0.5, partly, and 0 to No in order to produce quantitative results.

Furthermore, the work had to be graded on an average of 0.5. The principal author was in charge of applying the assessment parameter for quality to the studies, whereas the remaining authors were responsible for confirming the same assessments on a minority group of previously nominated studies. A few papers were omitted from the study using the same mechanism. However, as seen in Table 1, after going through the entire process, the final number of studies was 85. Any discrepancies were resolved through additional discussion. Any discrepancies were resolved through additional discussion.

## Data extraction

Counting the challenges and their practices is a difficult task because the majority of the challenges and practices from such areas are cascading in nature. Nonetheless, we discovered 11 primary challenges with 67 practices from a total of 85 papers. The identified challenges and their practices are presented in the next section via Tables 3–13, respectively. Furthermore, the papers extracted via the SLR are presented in (S1 Appendix) with the respective IDs and Title.

## Results and discussion

The goal of this research is to uncover the key issues and practices experienced by multiple stakeholders (such as clients and vendors) when dealing with COR.

The following are the challenges that we have identified and analyzed after thorough conduction of the SLR in such areas.

## Challenges in COR

Table 3 presents the 11 key challenges with the given frequencies as follows:

**1. Control and Coordination Challenges**:

'Control and Coordination' are one amongst the major challenge in COR. Control refers to a company's capability to command or manage scattered events so that they meet the

**Table 3. Challenges in COR.**

| S. NO | Challenges | Paper ID | Frequency |
|---|---|---|---|
| 1 | Control and Coordination Challenges | 3, 7 | 02 |
| 2 | Decision Problems | 2, 8, 5, 13,14, 28, 31, 33, 35, 40, 42, 54, 58, 61, 66, 71, 73, 76, 83, 84 | 20 |
| 3 | High Cost<br><br>(For example hidden cost, estimated-cost, control and coordination cost, contract cost etc.) | 3, 6, 7, 14, 17, 21, 28, 32, 35, 36, 39, 40, 42, 58,51, 66, 76 | 17 |
| 4 | Security and Compliance Challenges<br><br>(For example:<br><br>• Clouds Auditing<br><br> Managing the heterogeneity of services<br><br>• Coordinating the included organizations<br><br>• Client and vendor relationships management<br><br>• Localization as well as the migration of data and<br><br>• To cope with the lacking of security consciousness) | 8, 11, 20 | 03 |
| 5 | Risky Contracts | 3, 6, 25, 11, 14, 17, 18, 20, 21, 25, 27, 30, 35, 36, 39, 40, 41, 45, 56, 57, 60, 63, 85 | 23 |
| 6 | Poor Human Resource Management | 8, 11, 53, 59, 72 | 05 |
| 7 | IT Transitions Difficulties | 7, 14, 17, 21, 28, 31, 34, 39, 40, 42, 58 | 11 |
| 8 | Poor Contract Management | 1, 3, 5, 6, 7, 25, 8, 11, 12, 13, 14, 16, 18, 21, 23, 25, 29, 31, 34, 35, 36, 38, 40, 41, 45, 47, 50, 51, 52, 53, 57, 59, 62, 63, 65, 67, 68, 72, 76, 77, 84, 85 | 42 |
| 9 | Different Cultural Issues | 12, 35 | 02 |
| 10 | General Management Complexities | 35, 72 | 02 |
| 11 | Modern Technological Challenges | 35 | 01 |

company's goals. The ability of a company to coordinate these disparate activities is referred to as coordination [25].

**2. Decision Problems**:

Information system outsourcing decisions [26] are difficult to make because they entail a number of elements, including (a) establishing and managing a long-term partnership with an independent agent, and (b) revealing the critical organizational assets for controlling outside agents. When the same matter is evaluated from the perspective of COR, it becomes more challenging.

**Table 4. Practices to cope with control and coordination challenges in COR.**

| S.NO | Practices to cope with the control and coordination challenges |
|---|---|
| P#1.1 | Promote collaborative work environment |
| P#1.2 | Attain goal-oriented efforts |
| P#1.3 | Perform alliance-based activities |
| P#1.4 | Perform formally controlled mechanisms such as behavioral control etc. |
| P#1.5 | Encourage frequent communication |
| P#1.6 | Arrange review meetings on regular basis |
| P#1.7 | Build and maintain tasks completion checklists |

**Table 5. Practices to cope with decision problems in COR.**

| S.NO. | Practices to cope with decision problems |
|---|---|
| P#2.1 | Adopt simulation-based decision support system (DSS) framework such as GPS route planning etc. |
| P#2.2 | Adopt Multi Criteria Decision Making (MCDM) approaches such as AHP (Analytical Hierarchy Process) |
| P#2.3 | Negotiate the budget properly |

**Table 6. Practices to cope with high cost in COR.**

| S.NO. | Practices to cope with high cost |
|---|---|
| P#3.1 | Encourage alliance-based work |
| P#3.2 | Adopt Case Management approaches such as TCT (Transaction Cost Theory) |
| P#3.3 | Define the requirements clearly |
| P#3.4 | Use specialized analytics tools such as cost analysis etc. |
| P#3.5 | Establish and maintain a centralized repository |

**Table 7. Practices to cope with security and compliance challenges in COR.**

| S.NO. | Practices to cope with security and compliance challenges |
|---|---|
| P#4.1 | Use tailored software and in-house software such as Intrusion detection system etc. |
| P#4.2 | Adopt Layered Security Framework such as CSA Cloud Control Matrix (CCM) etc. |
| P#4.3 | Achieve high level of relevant certification such as CMMI etc. (for vendors) |
| P#4.4 | Choose certified vendors (for client) |
| P#4.5 | Adopt strong adherence systems |
| P#4.6 | Establish plans for disaster recovery |
| P#4.7 | Adopt standard frameworks like ISO 17799 etc. |
| P#4.8 | Establish threats identifications mechanisms, for example, appoint threat analyst etc. |
| P#4.9 | Encourage frequent communication |
| P#4.10 | Appoint skilled and highly qualified staff |
| P#4.11 | Arrange regular workshops for team awareness |
| P#4.12 | Provide opportunities for knowledge transfer such as group discussions etc. |
| P#4.13 | Adopt Service Level Agreements (SLA's)- (for vendor) |
| P#4.14 | Evaluate audit's reports and certifications of the vendors by the clients |
| P#4.15 | Use GRC tools such as Fusion Framework, IBM OpenPages etc. |

**Table 8. Practices to cope with risky contracts in COR.**

| S.NO. | Practices to cope with risky contracts |
|---|---|
| P#5.1 | Develop the contract efficiently and clearly |
| P#5.2 | Analyze the context clearly |
| P#5.3 | Encourage the compound approach |
| P#5.4 | Use appropriate tools such as probability and impact matrix etc. |
| P#5.5 | Follow standards such as ISO 37500 etc. |
| P#5.6 | Arrange review meeting on regular basis |
| P#5.7 | Appoint skilled and highly qualified staff |
| P#5.8 | Implement prototyping approaches |
| P#5.9 | Provide opportunities for knowledge transfer such as workshops etc. |
| P#5.10 | Encourage frequent communication |
| P#5.11 | Encourage alliance-based work |

**Table 9. Practices to cope with poor human resource management in COR.**

| S.NO. | Practices to cope with poor human resource management |
|---|---|
| P#6.1 | Use Human Resource Information System Software (HRIS) |
| P#6.2 | Engage the services of skilled staff |
| P#6.3 | Establish proper monitoring processes |
| P#6.4 | Promote the provision of strong online platform |

**Table 10. Practices to cope with IT transitions difficulties in COR.**

| S.NO. | Practices to cope with IT transitions difficulties |
|---|---|
| P#7.1 | Establish ontology-enabled semantic models such as Semantic Data Model (SDM) |
| P#7.2 | Establish and maintain a centralized repository |

**Table 11. Practices to cope with poor contract management in COR.**

| S.NO. | Practices to cope with poor contract management |
|---|---|
| P#8.1 | Promote penalties and Incentives |
| P#8.2 | Provide opportunities for knowledge transfer such as frequent discussions etc. |
| P#8.3 | Encourage long-lasting relationships |
| P#8.4 | Establish mutually dependent relationships |
| P#8.5 | Encourage frequent communication and coordination |
| P#8.6 | Promote integrity-based relationships |
| P#8.7 | Promote commitment and trust in relationships |
| P#8.8 | Encourage alliance-based relationships |
| P#8.9 | Engage the services of skilled staff |
| P#8.10 | Promote suitable governance-model |
| P#8.11 | Establish and maintain centralized repository |
| P#8.12 | Promote flexibility in contracts |
| P#8.13 | Achieve high level of relevant certification such as CMMI etc. (for vendors) |

**Table 12. Practices to cope with different cultural issues in COR.**

| S.NO. | Practices to cope with different cultural issues |
|---|---|
| P#9.1 | Prefer parties of similar culture, where possible |
| P#9.2 | Establish relationships with certified and trusted bodies |

**Table 13. Practices to cope with general management complexities in COR.**

| S.NO. | Practices to cope with general management complexities |
|---|---|
| P#10.1 | Encourage division of activities. i.e dividing an activity into subactivities, for example, plan, organize, target, motivate and control |
| P#10.2 | Promote commitment and trust in relationships |
| P#10.3 | Promote suitable governance-model |

**3. High Cost**:

The overall cost [27], for example, the hidden cost, estimated cost, control cost, coordination cost, etc., for outsourcing is high in the case of COR.

**4. Security and Compliance Challenges**:

Security and Compliance is a composite challenge, including six sub-challenges (see Table 3 above) in the context of COR. It is worth noting that this particular challenge focused on the cloud version of COR.

**5. Risky Contracts**:

Managing risk is also a major issue, specifically in the context of COR, where multiple stakeholders from different cultures and zones are involved. Furthermore, large or complex IT contracts, which are nearly always incomplete, enhance the chance of risk in a variety of ways.

**6. Poor Human Resource Management (HRM)**:

People and staff management becomes extremely difficult in the situation of COR, where multiple personnel from various cultural and zone variances are coordinated, particularly in a security-related system.

**7. IT Transitions Difficulties**:

One reason for this failure is the complication of information system outsourcing transitions. In the case of COR, where several clients and vendors are involved, the transaction will undoubtedly grow more complicated.

**8. Poor Contract Management**:

Contract management, also known as the hard side, is one of the biggest issues and is discussed in more detail in the literature. For pre-contract, the supplier's standard type contracts should not be utilized, even as a starting point, for complex outsourcing arrangements involving large sums of money, because they are always structured in favor of the outsourced vendor [28].

**9. Different Cultural Issues**:

Because of the increased level of uncertainty associated with the crossing of organisational, geographical and cultural boundaries, complex IS outsourcing strategies even become more complex [29].

**10. General Management Complexities**:

With several development centres in different time zones, geographies, and cultures, managing the complex outsourcing relationships is intrinsically tough. Furthermore, management becomes much more difficult when IT is mostly outsourced to one or more IT services providers i.e. COR.

**11. Modern Technological Challenges**:

Technology Challenges, Strategic Decision Challenges, Vendor Management Challenges, and Vendor Selection Challenges are the four key kinds of challenges often faced where more than one stackholders are involved [30].

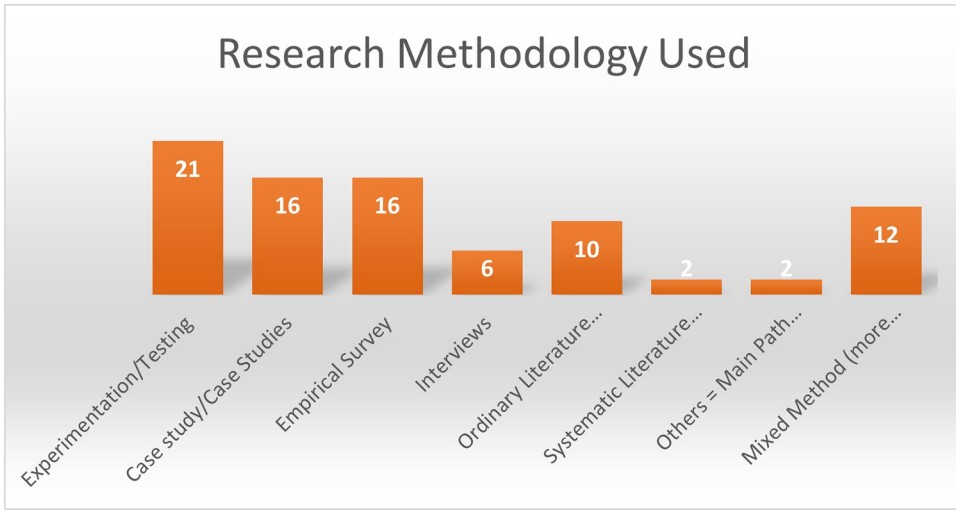

**Fig 2. Analysis based on research methodology.**

## Analysis of these challenges based on research methodology used

To analyze and well furnish the outcomes of the SLR, we further categorized the results of the identified challenges on 8 different commonly used research methodologies. The methodologies are Experimentation/Testing, Case Study/Case Studies, Empirical Survey, Interviews, Ordinary Literature Review (OLR), Systematic Literature Review (SLR), Others (i.e. Main Path Analysis and Content Analysis, Mixed Method (.i.e more than one method), as shown in (Fig 2). During the analysis, we concluded that Experimentation/Testing methodology has been used more in the literature in this domain. Because in these studies, models or methods were used to check or test the challenges quantitavely, for example, measuring the cost or taking decisions. Similarly, the second mostly used methodologies are Case Study/Case Studies or Empirical Survey. Once again the reason behind this analysis was to collect empirically sound data in the real context of a case. The Mixed Methodology shows progressive results in this particular area and that's why it is reported at number three position in the list etc.

## Analysis of these challenges based on publication period

Although, we did not put any date boundaries of data during the accomplishment of an SLR, but during the analysis, we pointed out a major contribution shift from a theoretical to practical implementation by the organizations in COR context. Therefore, based on this knowledge, we divided the publication era into two periods such as papers published before 2014 and papers published in 2014 and onwards, as shown in (Fig 3). We can further conclude from the analysis of (Fig 3) that COR is one of the most prominent research areas to date. Because sufficient number of studies are published in the second period (i.e., in 2014 and onwards).

## Analysis of these challenges based on publication venue

(Fig 4) gives an analysis based on publication channel. We divided the publication channels into three categories, such as papers published in Conference/Proceeding, Journal Papers, Technical Reports/Research Working Papers. From the analysis of (Fig 4), it is clearly visible that the majority of papers were published in Journals. This analysis once again is an evidence that COR is always a strong research area in the literature.

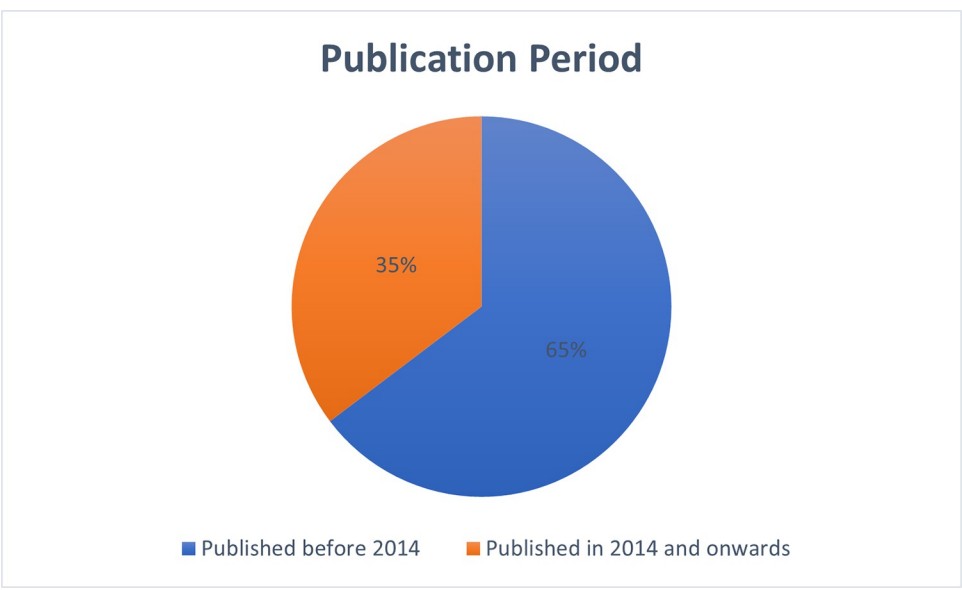

**Fig 3. Analysis based on publication period.**

### Practices for identified challenges

However, it was quite difficult to accurately account for the concerned practices for the stated challenges due to the cascading nature of data. Nonetheless, we organized them to the best of our efforts and identified 67 practices for the the aforementioned challenges. Moreover, due to the time and space constraints, we present these practices for the concerned challenges in tabular form, as depicted in Tables 4 to 14 below.

## Limitations and threats to validity

We did our best to conduct a systematic literature review as a study approach, including ensuring the pertinency of appropriate string selection and a sufficient sample size, but it is still

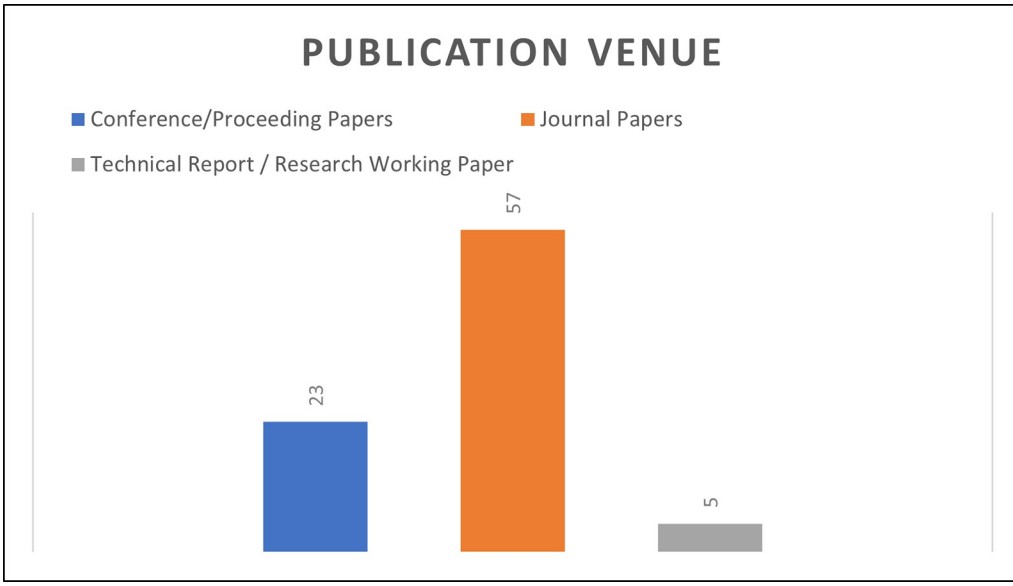

**Fig 4. Analysis based on publication venue.**

**Table 14. Practices to cope with modern technological challenges in COR.**

| S.NO. | Practices to cope with modern technological challenges |
|---|---|
| P#11.1 | Promote collaborative work environment |
| P#11.2 | Provide awareness opportunities to staff regarding the latest technologies |

conceivable that we have overlooked some crucial information. In order to reduce the danger of construct validity in terms of the usage of digital libraries, we narrowed our focus to pertinent and famous computing libraries, but again it was confined to the very few libraries. We have listed the publications in (S1 Appendix) for interested visitors to minimize the danger of the study's internal validity. To prevent such dangers, each component of our SLR was confirmed via a rigorous approach and intermittent review processes by the participating researchers. The identification method has been used in the literature for similar investigations multiple times. It should be noted that the majority of the studies we gathered were from the COR database. However, only a few of them, or portions of them, do not properly identify the COR data, giving the context a hazy picture. Furthermore, only papers written in English were chosen.

## Conclusion

The benefits of a complex IT outsourcing arrangement include enhanced cost certainty and lower expenses, increased scalability, and flexibility on demand. Complex outsourcing, on the other hand, demands interactions between many stakeholders from various areas and cultures, making it substantially more difficult to manage than regular outsourcing. Complex outsourcing is also more challenging than other forms of outsourcing since it needs a variety of control and coordination mechanisms for project management, which increases the chance of project failure accordingly. In this regard, a more systematic research is needed to uncover the important difficulties and techniques in this area in order to overcome project failure in complex outsourcing relationships. As a result, this study uses systematic literature review as a research method and serves as a pioneering effort to achieve the aforementioned goals. The authors highlighted 11 key issues after receiving the SLR data, with 67 practises in hand from a total of 85 publications. The authors want to build a comprehensive framework based on their findings in the future by using some strong approaches like as AHP and fuzzy logic, etc.

## Supporting information

**S1 Appendix.**
(DOCX)

**S1 File. Authors' biography.**
(DOCX)

## Acknowledgments

We are grateful to other members of the Software Engineering Research Group (SERG) for their valuable feedback and support during the research process.

## Author Contributions

**Conceptualization:** Habib Ullah Khan, Muhammad Ilyas.

**Data curation:** Habib Ullah Khan.

**Formal analysis:** Ghulam Murtaza Khan, Habib Ullah Khan.

**Methodology:** Ghulam Murtaza Khan, Siffat Ullah Khan.

**Project administration:** Siffat Ullah Khan, Muhammad Ilyas.

**Resources:** Muhammad Ilyas.

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
