## [Decision Letter · Decision Letter 0]

15 Nov 2021

PONE-D-21-28732Challenges And Practices Identification In Complex Outsourcing Relationship: A Systematic Literature ReviewPLOS ONE

Dear Dr. Khan,

Thank you for submitting your manuscript to PLOS ONE. After careful consideration, we feel that it has merit but does not fully meet PLOS ONE’s publication criteria as it currently stands. Therefore, we invite you to submit a revised version of the manuscript that addresses the points raised during the review process.

We look forward to receiving your revised manuscript.

Kind regards,

Anandakumar Haldorai, PhD

Academic Editor

PLOS ONE

Journal Requirements:

2. Please amend your Methods section to include the date range for the search. Please also state the latest date on which the search was performed.

5. Please update your submission to use the PLOS LaTeX template. The template and more information on our requirements for LaTeX submissions can be found at http://journals.plos.org/plosone/s/latex.

Additional Editor Comments:

Please carefully address the issues raised in the comments and, up front in your revised paper. Your revised paper will be sent to the same reviewers, as well as possibly new reviewers, for evaluation.

Make sure the Abstract briefly describes the paper as it is used in abstracting and citation services. Keep the Abstract between 200 words. Do not use any references in the Abstract.

Spell out each acronym the first time used in the body of the paper. Spell out acronyms in the Abstract only if used there.

Include a list of six to ten key words after the Abstract.

You may ignore any suggestion of including self-references by reviewers if not applicable.

Include a paragraph at the end of the Introduction describing the organization of the paper.

Make sure that the Conclusion briefly summarizes the results of the paper it should not repeat phrases from the Introduction. Keep the Conclusion to about 300 words. Do not use any references or acronyms in the Conclusion.

Make sure all figures and tables are referred to in the body of the paper.

It is recommended to use a professional native English-speaking editor. Papers with less than excellent English will not be published even if technically perfect.

Reviewers' comments:

Reviewer's Responses to Questions

**Comments to the Author**

1. Is the manuscript technically sound, and do the data support the conclusions?

Reviewer #1: Partly

Reviewer #2: Yes

2. Has the statistical analysis been performed appropriately and rigorously? 

Reviewer #1: Yes

Reviewer #2: Yes

3. Have the authors made all data underlying the findings in their manuscript fully available?

Reviewer #1: No

Reviewer #2: Yes

4. Is the manuscript presented in an intelligible fashion and written in standard English?

Reviewer #1: Yes

Reviewer #2: Yes

5. Review Comments to the Author

Reviewer #1: The structure of the paper is good, however, it needs the following modifications to be published in the journal.

1. Abstract need to be revised to specify the demand for the review.

2. In introduction part, authors should provide more priority about outsourcing relationship.

3. The methodology of the proposed review is complex and readers find problems in understanding the procedure.

4. There is a need for more research papers that present more detailed information about outsourcing relationships and its advantages.

5. Finally, the conclusion part need a rigorous revision.

Reviewer #2: It’s a great work and great sense of methodology and great analyzing and I wish to see more relate study and more upgrades with the same concept of research in the future and keep it up and good work

6. PLOS authors have the option to publish the peer review history of their article (what does this mean?). If published, this will include your full peer review and any attached files.

Reviewer #1: No

Reviewer #2: **Yes: **Haitham Medhat Abdelaziz Elsayed Aboulilah

---

## [Author Response · Author response to Decision Letter 0]

21 Dec 2021

Rebuttal Letter

Original Manuscript ID: PONE-D-21-28732

Original Article Title: “Challenges And Practices Identification In Complex Outsourcing Relationship: A Systematic Literature Review”

To: PLOS ONE

Re: Response to reviewers

Dear Editor,

Thank you for allowing a resubmission of our manuscript, with an opportunity to address the reviewers’ comments.

We are uploading 

(1) Our point-to-point response to the comments below (response to reviewer(s) and academic editor, respectively) 

(2) A marked-up coy with track changes (Revised Manuscript with Track Changes)

 (3) An unmarked version without track changes (Manuscript)

Best regards,

<author name> et al.

Reviewer#1, Concern # 1: 

Reviewer’s comment: Abstract need to be revised to specify the demand for the review

Authors’ response: Many thanks for your kind suggestion. We have revised the abstract by adding such specific demand.________________________________________

Reviewer#1, Concern # 2: 

Reviewer’s comment: In introduction part, authors should provide more priority about outsourcing relationship.

Authors’ response: We have added few more papers having specific focus on outsourcing relationships in the introduction section of the manuscript.________________________________________

Reviewer#1, Concern # 3: 

Reviewer’s comment: The methodology of the proposed review is complex and readers find problems in understanding the procedure.

Authors’ response: We have performed systematic literature review (SLR) as a research methodology. We have followed the SLR guidelines from [25, 26], respectively. SLR is widely used research methodology in software engineering and we have used this method in our previous work having Doi as follows: 10.1109/ACCESS.2021.3085707, 10.1016/j.infsof.2010.08.003, and 10.1109/ICGSE.2009.28 etc. ________________________________________

Reviewer#1, Concern # 4: 

Reviewer’s comment: There is a need for more research papers that present more detailed information about outsourcing relationships and its advantages.

Authors’ response: Our search is based on our pre-defined search string that was constructed during the planning phase of the SLR, i.e., SLR protocol. During the implementation phase of the SLR protocol, the search phase was completed in April, 2021. Now, according to this concern, we have added some more latest papers, through snowballing technique, relevant to outsourcing relationships. 

Reviewer#1, Concern # 5: 

Reviewer’s comment: Finally, the conclusion part need a rigorous revision.

Authors’ response: We have revised the whole manuscript specifically the conclusion section. ________________________________________

NOTE: We have further addressed the Journal Requirements as follows:

Concern # 1:

Authors’ response: We have followed the PLOS ONE’S style and format in the revised manuscript.

Concern # 2:

2. Please amend your Methods section to include the date range for the search. Please also state the latest date on which the search was performed.

Authors’ response: We have amended the Methods section, accordingly. Please see the change tracked at page 8 under the Literature Resources section of Research Methodology. During the implementation phase of the SLR protocol, the search phase was completed in April, 2021. 

Concern # 3:

Authors’ response: No fund has been issued to us by any organization/agency. It is solely accomplished by the authors. However, the publication charges are sponsored by the Qatar university. 

Concern # 4:

Authors’ response: ?

Concern # 5:

5. Please update your submission to use the PLOS LaTeX template. The template and more information on our requirements for LaTeX submissions can be found at http://journals.plos.org/plosone/s/latex.

Authors’ response: We have updated the manuscript according to the PLOS LaTeX template, by following the given link.

Concern # 6:

Authors’ response: We have reviewed the references list and revised it properly, and ensured its correctness and completeness. We have added the references of additional papers which we have included now, as according to reviewers’ suggestions. Furthermore, we have removed/retracted the references of excluded papers, because these references have been removed during merging and updating the table data for further improvements. 

Additional Editor Comments:

Concerns #:

- Please carefully address the issues raised in the comments and, up front in your revised paper. Your revised paper will be sent to the same reviewers, as well as possibly new reviewers, for evaluation.

- Make sure the Abstract briefly describes the paper as it is used in abstracting and citation services. Keep the Abstract between 200 words. Do not use any references in the Abstract.

- Spell out each acronym the first time used in the body of the paper. Spell out acronyms in the Abstract only if used there.

- Include a list of six to ten key words after the Abstract.

- You may ignore any suggestion of including self-references by reviewers if not applicable.

- Include a paragraph at the end of the Introduction describing the organization of the paper.

- Make sure that the Conclusion briefly summarizes the results of the paper it should not repeat phrases from the Introduction. Keep the Conclusion to about 300 words.

- Do not use any references or acronyms in the Conclusion.

- Make sure all figures and tables are referred to in the body of the paper.

- It is recommended to use a professional native English-speaking editor. Papers with less than excellent English will not be published even if technically perfect.

Authors’ response: We have addressed all the mentioned comments/concerns, accordingly.

NOTE: It is worth noting that we have revised or renamed few challenges, and similarly, merged together some of the previously identified practices for improvements. In this way, we finally collected 67 practices for 11 identified challenges.

---

## [Decision Letter · Decision Letter 1]

3 Jan 2022

Challenges And Practices Identification In Complex Outsourcing Relationship: A Systematic Literature Review

PONE-D-21-28732R1

Dear Dr. Khan,

We’re pleased to inform you that your manuscript has been judged scientifically suitable for publication and will be formally accepted for publication once it meets all outstanding technical requirements.

Kind regards,

Anandakumar Haldorai, PhD

Academic Editor

PLOS ONE

Reviewers' comments:

Reviewer's Responses to Questions

**Comments to the Author**

1. If the authors have adequately addressed your comments raised in a previous round of review and you feel that this manuscript is now acceptable for publication, you may indicate that here to bypass the “Comments to the Author” section, enter your conflict of interest statement in the “Confidential to Editor” section, and submit your "Accept" recommendation.

Reviewer #1: All comments have been addressed

Reviewer #2: All comments have been addressed

2. Is the manuscript technically sound, and do the data support the conclusions?

Reviewer #1: Yes

Reviewer #2: Yes

3. Has the statistical analysis been performed appropriately and rigorously? 

Reviewer #1: Yes

Reviewer #2: Yes

4. Have the authors made all data underlying the findings in their manuscript fully available?

Reviewer #1: Yes

Reviewer #2: Yes

5. Is the manuscript presented in an intelligible fashion and written in standard English?

Reviewer #1: Yes

Reviewer #2: Yes

6. Review Comments to the Author

Reviewer #1: The paper is acceptable in its current form. No need to modify. The author had addressed the comments

Reviewer #2: It’s nice research paper and really good analysis with a perfect methodology and based on good theoretical and looking forward to see more papers can doing with same content and research base as same as the current one cause really it’s open for opportunity to creat a new one , wish you all the best

7. PLOS authors have the option to publish the peer review history of their article (what does this mean?). If published, this will include your full peer review and any attached files.

Reviewer #1: No

Reviewer #2: **Yes: **Haitham Medhat Abdelaziz Elsayed Aboulilah

---

## [Editor Report · Acceptance letter]

17 Jan 2022

PONE-D-21-28732R1 

Challenges and practices identification in complex outsourcing relationships: A systematic literature review 

Dear Dr. Khan:

I'm pleased to inform you that your manuscript has been deemed suitable for publication in PLOS ONE. Congratulations! Your manuscript is now with our production department. 

Kind regards, 

on behalf of

Dr. Anandakumar Haldorai 

Academic Editor

PLOS ONE